# TPC-NAS: Sub-Five-Minute Neural Architecture Search for Image Classification, Object-Detection, and Super-Resolution

## Abstract

Neural network models have become more sophisticated with the explosive development of AI and its applications. Automating the model search process is essential to explore a full range of neural architectures for satisfactory performance. However, most current NAS algorithms consume significant time and computing resources, and many cater only to image classification applications. This paper proposes the total path count (TPC) score, which requires only simple calculation based on the architecture information, as an efficient accuracy predictor. TPC score is not only simple to come by but also very effective. The Kendall rank correlation coefficient of the TPC scores and the accuracies of 20 architectures for the CIFAR100 problem is as high as 0.87. This paper also proposes TPC-NAS, a zero-shot NAS method leveraging the novel TPC score. TPC-NAS requires no training and inference, and can complete a NAS task for Imagenet and other vision applications in less than five CPU minutes. Then, we apply TPC-NAS to image classification, object detection, and super-resolution applications for further validation. In image classification, TPC-NAS finds an architecture that achieves 76.4% top-1 accuracy in ImageNet with 355M FLOPs, outperforming other NAS solutions. Starting with yolov4-p5, TPC-NAS comes up with a high-performance architecture with at least 2% mAP improvement over other NAS algorithms' results in object detection. Finally, in the super-resolution application, TPC-NAS discovers an architecture with fewer than 300K parameters and generates images with 32.09dB PSNR in the Urban100 dataset. These three experiments convince us that the TPC-NAS method can swiftly deliver high-quality CNN architectures in diverse applications. The related source code is available at `https://github.com/TPC-NAS/TPC`.

## 1 Introduction

The complexity of high-performance machine learning models has skyrocketed, and manual tuning of hyperparameters and neural network (NN) architecture has become laborious and time-consuming. More efficient methodologies for the design, training, and deployment of NN models are required. Toward this end, recently, we have witnessed rapid growth in research on neural (network) architecture search (NAS) that automates the model search process.

Early NAS algorithms use evolutionary search (Real et al., 2017; 2019) or reinforcement learning (Zoph & Le, 2017; Tan et al., 2019). However, such methods typically require multiple training of different architectures, which consumes significant computational resources and time. To reduce search time, differentiable NAS employs gradient descent (Mei et al., 2020; Chen et al., 2021b; Xu et al., 2020) that decides which architectures to keep by updating the weights between different operations. DARTS (Liu et al., 2019), for example, makes the search space continuous by applying a softmax function to all possible operations. After training, only the operations with the highest softmax output will be retained as the final searched model. Later, Wang et al. (2021c) discovered that deciding on the final model based on its contribution to supernet performance outperforms deciding on the final model solely on the softmax output between architectures. Although gradient-based algorithms speed up the search process, they require the construction of a supernet that can

cover all search spaces, which typically involves a large amount of memory, making this method unsuitable for large and complex problems.

Around the same time as the gradient-descent-based methods were developed, one-shot NAS methods (Guo et al., 2020; Cai et al., 2020; Zela et al., 2020; Stamoulis et al., 2019) were proposed. In contrast to the gradient-descent-based methods that require training the overall supernet once, the one-shot methods typically have two steps: training and searching. The one-shot methods applied the weight-sharing technique in training, thus significantly reducing the number of times the model needs to be trained. Furthermore, the one-shot methods only sample and train one subnet from the supernet at a time. For example, Wang et al. (2021b) samples the model with the best or the worst performance to improve the supernet's overall performance. To ensure that individual models can be trained fairly, Chu et al. (2021b) proposed that all architectures should be sampled equally. Since only one subnet's data are stored at a time, the one-shot methods have better memory usage. During the search process, the one-shot methods set the hardware constraints and select the subnet that achieves the highest performance while meeting those constraints. However, the subnets are interconnected, and it is difficult to ensure that a single subnet can be trained appropriately. Although few-shot methods (Hu et al., 2022; Zhao et al., 2021) effectively mitigate this problem by dividing a large supernet into several smaller sub-supernets, it is still difficult to ensure that the sampled subnet with the highest accuracy will still perform as expected when trained separately.

In addition to the shortcomings above, most NAS algorithms share a common flaw: it takes too much time and memory to complete the architecture search. This daunting requirement on computing resources often poses a high barrier to entry for the average NN users. Hence, this paper proposes a novel zero-shot NAS algorithm with an accuracy predictor based on a neural network's total path count (TPC) between the first layer's input nodes and the final layer's output nodes. The more paths a NN model has, the greater the expressive power of the NN model to perform different tasks and achieve higher accuracy. Most importantly, the TPC score is determined solely by the NN structure, and no weight training of models is required, which significantly reduces the search complexity. TPC scores can be computed in as little as 10 microseconds of CPU time and correlate well with the NN architecture performances. As a result, the proposed TPC-NAS method, which uses the TPC score in the standard zero-shot NAS search, can complete a NAS task in a matter of minutes.

Many previous NAS researches validated their approaches with a few image classification tasks. Whether or not their proposed NAS solutions will generalize well in other applications remains to be assessed. Furthermore, building and training a supernet from scratch takes enormous effort, limiting the feasibility of a broader range of applications. Toward this end, TPC-NAS has been applied to image classification, object detection, and super-resolution applications and has achieved overwhelming success. In all three applications, the architectures found by TPC-NAS outperform all the manually-designed and most NAS-based architectures with comparable complexity.

Our contributions are summarized as follows:

1. We propose the TPC score, a simple yet effective accuracy predictor. This score requires only the knowledge of a model's structure parameters to predict its expressivity and performance; thus, the score computation time is a few microseconds.

2. The TPC score correlates very well with the NN model's accuracy. The TPC-based zero-shot NAS algorithm we propose can be implemented on CPUs or edge devices. Typical search times are within five minutes on CPU.

3. TPC-NAS is the first zero-shot NAS algorithm applied to image classification, object detection, and super-resolution. TPC-NAS can swiftly find architectures outperforming the hand-crafted and NAS-discovered architectures in all three applications.

This paper is organized as follows. Section 2 discusses related works in the field of NAS. The principle of our TPC score and the TPC-NAS algorithm are described in detail in Section 3. Section 4 explains how we apply TPC-NAS to image classification, object detection, and super-resolution and shows the experimental results. Then, several issues are discussed in Section 5, and Section 6 concludes this paper.

## 2 RELATED WORKS

### 2.1 NEURAL ARCHITECTURE SEARCH FOR CNNS

Many NAS algorithms have been proposed over the years. Among them, zero-shot NAS algorithms (Lin et al., 2021; Chen et al., 2021a; Mellor et al., 2021; Shu et al., 2022) do not optimize model weights and instead explore only the hyperparameters, thus significantly speeding up the search process. The majority of zero-shot algorithms consist of a performance predictor and a genetic algorithm for architecture search. The performance predictor usually determines the associated NAS algorithm's execution time and the quality of the architecture found. For example, Lin et al. (2021) predicts model performance by calculating the variances of the BN layers and observing how data spread across layers. Chen et al. (2021a) found it is natural to measure the expressivity of a ReLU network with the number of linear regions it can separate. Mellor et al. (2021) empirically find that the correlation between sample-wise input-output Jacobian can indicate the architecture's test performance. Shu et al. (2022) discovered that the Neural Tangent Kernel (NTK) is a sound performance predictor because it can characterize the CNN performance at initialization.

Although the accuracy predictors mentioned above can predict the performance of the architecture without training it, they still need to perform the forward or backward pass of the model when evaluating the architecture, which also makes their search time at least several hours. As a result, we propose our TPC-NAS, which requires only architecture information to predict model performance and thus significantly reduces our search time to less than 5 minutes.

As for the genetic algorithm for architecture search (Reeves, 2010), there is usually an initial population of architectures, and individuals (architectures) with higher scores can be found by mutation and cross-over operations on the current population. In various genetic algorithms, natural selection mechanism can be quite different. Typically, natural selection is achieved by pruning the individual with the lowest score from the population. Still, some study (Ghosh et al., 1996) stipulated that removing the eldest individual can achieve better outcomes.

### 2.2 TASK-AGNOSTIC NEURAL ARCHITECTURE SEARCH

Early NAS algorithms almost always adopted image classification problems to demonstrate the validity of their solutions. Despite many good results in image classification, we could not be assured that NAS algorithms can be extended to search architectures for other applications. Afterwards, researchers attempted to apply NAS to other vision applications, such as super-resolution (Song et al., 2020; Chu et al., 2021a; 2020) or object detection (Ghiasi et al., 2019; Wang et al., 2022; Yao et al., 2020).

Unfortunately, almost all NAS works applied their NAS methods to finding models for a single task. The underlying reasons for this limitation are as follows. First, using a complex NAS algorithm to search models for a new problem usually requires significant effort. Many researchers may not have enough experience or time to extend their solutions to other tasks. Second, there exist substantial structural differences between the models for different applications. For example, the models for the super-resolution task (Liu et al., 2020; Ahn et al., 2018) usually need many skip connections or concatenation layers to retain most information of the input images. On the other hand, the object-detection models (Bochkovskiy et al., 2020) usually need multiple detection heads to detect objects of different sizes. The differences in architectures for various applications may make it difficult for NAS designers to find a single generalized solution that applies to all types of models.

Toward this end, this work proposes a simple and general predictor suitable for predicting the performance of CNN models for vision applications. Furthermore, the proposed TPC-NAS solution has been successfully applied to discover high-performance models for the following three problems: image classification, object detection, and super-resolution.

## 3 EXPRESSIVITY AND TPC-NAS

This section first introduces the TPC score and demonstrates how it works. Then, we show the Kendall rank correlation coefficient between the test accuracy and the TPC score of several models

to illustrate the effectiveness of the TPC score. Finally, the TPC-based zero-shot NAS algorithm is presented.

### 3.1 EXPRESSIVITY OF VANILLA CONVOLUTIONAL NEURAL NETWORKS

Expressivity (Gühring et al., 2020) aims to describe an architecture's ability to tackle a variety of problems or datasets and is used to evaluate the quality of the architectures in some zero-shot NAS algorithms. For example, simple architectures with only a few hidden layers and parameters can only implement a limited number of transformations on input images. As a result, this model cannot fully extract/capture the input image's information, and the model's expressivity is relatively poor.

However, expressivity is now more of a concept, and there are numerous ways to estimate it. For example, Zen-NAS describes the expressivity by getting the variance in the BN layer and observing how the data spread across layers. Most methods for calculating expressivity remove skip connections, concatenations, and other complex structures because they do not make a huge difference in expressivity. With this simplification, a model retains only the main skeleton, including convolutional layers, full-connected layers, and ReLU, which is the so-called "vanilla" convolutional neural network (VCNN) (Lin et al., 2021). In this paper, we propose to estimate the expressivity of a VCNN model with the total number of paths from the input layer to the output layer of that model.

### 3.2 TPC SCORE

For a vanilla convolutional neural network, we can view the whole model as a huge graph. Neurons of each layer can be regarded as nodes, connected by outgoing edges that carry different weight values. In such a directed graph, many paths exist between the input nodes of the first layer and the output nodes of the last layer. It is conceivable that with more paths in a graph, the corresponding VCNN can apply more transformations to the inputs and therefore has a higher expressivity.

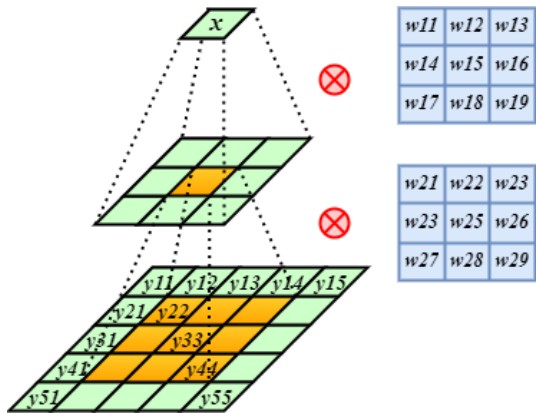

Take Figure. 1 as an example. There is only one input node $x$, and the first and the second convolutional layers both have 3x3 convolutional kernels. As the signal traverses the layers in the graph (model), input information is passed on to the ensuing layers via the weights on the directed edges. The information accumulated on the output pixel $y_{i,j}$ is given by

Figure 1: A simple VCNN model. In each of the two convolutional layers, one input pixel can affect nine output pixels. Overall, there are 81 different paths in the model from the input layer to the output layer, and the TPC score is 81.

$$y_{i,j} = \sum_{(k,l) \in A_{i,j}} x * w_{1k} * w_{2l}, \tag{1}$$

where $A_{i,j}$ represents the collection of paths that end at output pixel $y_{i,j}$ and $(k,l)$ denotes a path through the $k$th and the $l$th weights in the first and second kernels, respectively.

A VCNN's expressivity should be related to all its input and output nodes. Therefore, we adopt the number of paths of the VCNN as an expressivity measure, which is formulated as

$$\text{TPC} = \sum_{i,j} |A_{i,j}|. \tag{2}$$

In the example of Fig. 1, there are 81 paths.

To simplify the path number calculation of deeper vanilla convolutional neural networks with more layers, we calculate the path number of the entire graph layer by layer. We can recursively obtain the path number of the subgraph up to the $n+1$th layer by multiplying the path number of the $n$-layer subgraph with the number of ways that a path ending in the $n$th layer can extend to the $n+1$th layer.

The number of possible extending paths from a node is called *outdegree* in graph theory. Taking the convolutional layer as an example, we can compute the outdegree of its layer $p$ ($O_p$) as

$$O_p = \frac{C_{o,p} * k_p^2}{q_p^2 * g_p}, \tag{3}$$

where $C_{o,p}$, $k_p$, $q_p$, and $g_p$ stand for the number of output channels, the kernel size, the stride, and the number of groups of layer $p$, respectively. However, the actual number of total paths in a VCNN is usually large. To reduce its magnitude and for ease of computation, we take the log of the total number of paths. So, we propose the following TPC score ($S_t$) for a VCNN

$$S_t = \sum_p \log(O_p). \tag{4}$$

Algorithm 1 then lists the procedure of how the TPC score is calculated. For a given NN architecture, we first remove the skip connections and keep only the backbone as the VCNN. Then, Eq. (4) is applied to the VCNN to obtain the TPC score.

---

**Algorithm 1** TPC Score

**Require:** $C_{o,p}$, $k_p$, $q_p$ and $g_p$ for all CONV layers
**Ensure:** TPC
  1: $S_t = 0$
  2: Remove all residual links in the model
  3: For each CONV layer $p$ do:
     $S_t$ += $\log\left(\frac{C_{o,p}*k_p^2}{q_p^2*g_p}\right)$
  4: **return** $S_t$

---

### 3.3 TPC Score of Separable Convolution Layer

Consider the separable CONV layer in Figure 2 as a TPC score calculation example. A separable CONV layer can be divided into two distinct CONV layers: depthwise CONV and pointwise CONV. An input pixel is connected to $k^2$ output pixels in the depthwise CONV layer, and an input pixel in the pointwise CONV layer is connected to $C_o$ output pixels. Take the depthwise and pointwise layers together, and there are $C_o * k^2$ paths from an input of the separable

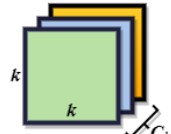

**Depthwise CONV weight    Pointwise CONV weight**

Figure 2: Illustration of the TPC score for the separable CONV layer.

CONV layer to its output pixels, which is identical to the TPC score of the corresponding conventional CONV layer. The above TPC score outcome is in line with the expectation that the separable CONV layer achieves the same expressivity as the corresponding conventional CONV layer.

### 3.4 Kendall's $\tau$-score

We conducted a further investigation to evaluate how good the proposed TPC score is as an accuracy predictor of CNN architectures. We randomly selected 20 different architectures from the search space, trained the architecture on the CIFAR100 dataset, and found the test accuracy. These 20 test accuracies and the corresponding TPC scores are plotted in Figure 3. Clearly, the TPC score has a strong positive correlation with test accuracy. In addition, we computed the Kendall rank correlation coefficient, which was 0.87, on par with other accuracy predictors (Lin et al., 2021).

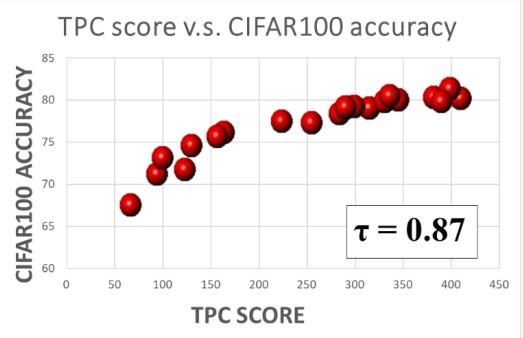

Figure 3: TPC score v.s. CIFAR100 accuracy. For this evaluation, we randomly sampled 20 structures from the Resnet search space (Appendix A.1), which are the same as those used in Zen-NAS work (Lin et al., 2021).

### 3.5 TPC-NAS Algorithm

Based on the TPC score, we introduce the TPC-NAS, a zero-shot NAS algorithm using an evolutional search. The TPC-NAS algorithm is described in Algorithm 2. In Appendix A.1, we have delineated in more detail the search spaces adopted in three different applications. The initial model $F_0$ is typically set as a small architecture in the search space. The MUTATE function creates new models from the search space according to Algorithm 3. A new model is evaluated to see if it meets the hardware constraints. If it does, its TPC score will be calculated, and it will be inserted into the population. When the total number of models in the population exceeds the predefined upper bound, the model with the lowest TPC score will be deleted. At the end of the iteration process, the model with the highest TPC score will be the outcome of TPC-NAS.

---

**Algorithm 2** TPC-NAS Algorithm

---

**Require:** search space $\mathcal{S}$, hardware constraints $K$, maximal depth $D$, number of iterations $M$, population size $N$, initial structure $F_0$
1: Initial population $\mathcal{P} = \{F_0\}$
2: **for** $m = 1, 2, \ldots, M$ **do**
3:     Randomly select one model $F$ from the population.
4:     Structure $\hat{F} = \text{MUTATE}(F, \mathcal{S})$
5:     **if** $\hat{F}$ does not satisfy hardware constraints $K$ or has more than $D$ layers **then**
6:         Go to line 13
7:     **else**
8:         Calculate $S_t(\hat{F})$
9:         Insert $\hat{F}$ to population $\mathcal{P}$
10:     **end if**
11:     Remove $F_{\min}$, the model with the lowest $S_t$, if $|\mathcal{P}|$ exceeds $N$
12: **end for**
13: **return** $F_{\max}$, the model with the highest $S_t$

---

**Algorithm 3** MUTATE

---

**Require:** Structure $F$, search space $\mathcal{S}$
1: Randomly select a block $b$ in structure $F$
2: Randomly changes the block type, kernel size, width and depth of $b$ within search space $\mathcal{S}$
3: **return** Mutated structure $\hat{F}$

---

## 4 Experiments

The proposed TPC-NAS method was applied to several vision applications to demonstrate its effectiveness. We adopted three complicated applications: image classification, object detection, and super-resolution. In this section, we will show that TPC-NAS indeed can always find high-performance CNN models for complicated tasks in only a few minutes.

### 4.1 TPC-NAS for Image Classification CNN

We first adopted the evolutionary genetic algorithm to find the best architecture for the CIFAR10 and CIFAR100 datasets, using several zero-shot proxies with a parameter limit of 1M and 1440 epochs of training. In this experiment, we adopted the search space and training mechanism as in the Zen-NAS work (Lin et al., 2021). All NAS methods are different only in the accuracy prediction scores used. Table 1 shows that the model found with the TPC score outperforms other zero-shot NAS methods in terms of accuracy.

Most zero-shot algorithms require running the model's forward or backward process to calculate the scores. For example, NASWOT must go backward to obtain gradient information to compute its score. Zen-NAS only needs to perform the forward process once to compute the Zen score from the data in the BN layers. On the contrary, TPC score calculation does not require forward or backward propagation. Only the structural parameters of the architecture are sufficient. The times specified in Table I list how long different zero-shot scores take to compute for a ResNet-18 at 224*224 input resolution. The Zen score, which requires inference once, is faster than NASWOT and TE-Score. In comparison, the TPC score skips the inference step and is nearly 1000 times faster than Zen score.

Table 1: CIFAR10 and CIFAR100 results for different zero-shot NAS methods under 1M model size constraint. All data except the TPC-NAS data are from (Lin et al., 2021). All other hyperparameters, such as population size and learning rate, are identical. The "Time" column represents the time to compute the scores of ResNet-18 model with 224*224 input resolution.

|  | CIFAR10 | CIFAR100 | Time (sec) |
| --- | --- | --- | --- |
| TPC score | **97.1 %** | **81.1 %** | 0.00001 |
| Zen-Score (Lin et al., 2021) | 96.2 % | 80.1 % | 0.012 |
| TE-Score (Chen et al., 2021a) | 96.1 % | 77.2 % | 0.34 |
| NASWOT (Mellor et al., 2021) | 96.0 % | 77.5 % | 0.04 |
| Synflow (Tanaka et al., 2020) | 95.1 % | 75.9 % | 0.04 |

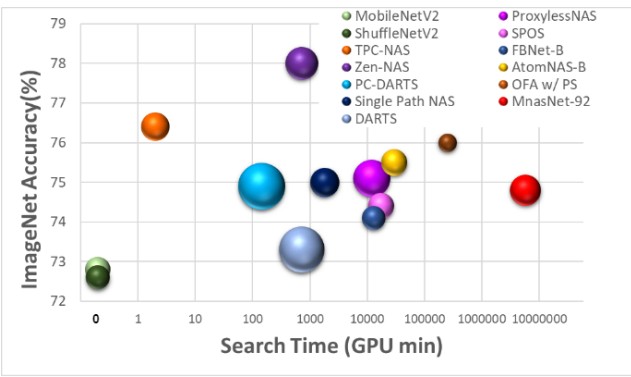

Figure 4: ImageNet accuracy for different NAS algorithms or manually-designed models. The x-coordinate is the search time on GPU, the y-coordinate is the model's accuracy on ImageNet, and the bubble size represents the model's FLOPs. The complete comparison table is in Appendix A.2.

We then experimented with the ImageNet classification problem using the TPC-NAS algorithm and the search is provided in A.1. Fig. 4 compares TPC-NAS with other NAS algorithms and hand-crafted models. The size of the balls represents the FLOPs of the model used to run the ImageNet classification. Note that all hand-crafted models, such as MobileNetV2 and ShuffleNetV2, require zero search time. First, all models found by NAS algorithms achieve better performance than the models designed by humans, demonstrating the power of automation. Secondly, TPC-NAS (orange bubble) delivers a model achieving 76.4% ImageNet Top-1 accuracy with 355M FLOPs, outperforming most NAS algorithms. Only one method, i.e., Zen-NAS, finds a model with 78.0% accuracy. But the required Zen-NAS search time on GPU is more than 350 times the TPC-NAS search time on CPU. Also, the model found by Zen-NAS is 15% larger in terms of FLOPs. All things considered, TPC-NAS represents a very competitive NAS method for image classification applications.

## 4.2 TPC-NAS FOR OBJECT DETECTION CNN

We also applied the TPC-NAS algorithm to search CNNs for object detection using the COCO-2017 dataset. The initial architecture in the search was the YOLOv4-p5 (Wang et al., 2021a) architecture, which has an obvious backbone and head. The backbone makes up most of the computation and is typically built with an image classification architecture. In this experiment, we mainly stuck with the YOLOv4-p5 main architecture and only changed structural parameters such as the number of layers, kernel size, etc. The NAS objective in this application is to find a smaller version of YOLOv4-p5 under the given hardware constraints, 20G and 40G FLOPs. Starting from a well-known and high-performance model, TPC-NAS can quickly improve the efficiency of the solution by finding a smaller yet accurate model.

Figure. 5 illustrates the TPC-NAS search results and compares them with several famous NAS methods and manually-designed models. TPC-NAS requires only 3 minutes to find a satisfactory model, which is over 1000 times faster than other NAS algorithms. Furthermore, the TPC-YOLO models

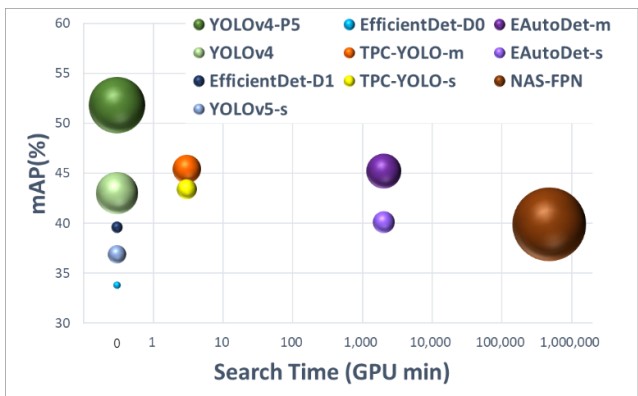

Figure 5: Object Detection results for different NSA algorithms or manually-designed models. The x-coordinate is the search time on GPU (except for TPC-NAS), the y-coordinate is the mAP (mean average precision) of the COCO dataset, and the bubble size represents each model's FLOPs. Models designed by humans have no search time. The complete comparison table is in Appendix A.3.

found achieves 43.4% and 45.4% mAP under 20G and 40G FLOPs, respectively. This means that the TPC-NAS model's mean average precision performance is higher than other architectures with similar complexities (bubble sizes). The only architecture that outperforms ours is the YOLOv4-p5 model, which has 164G FLOPs, four times higher than the TOC-NAS model. Despite exploring only some structural parameters of a known model, the TPC-NAS method can effectively downsize the FLOPs and arrive at a competitive solution in a very short interval.

### 4.3 TPC-NAS for Super-resolution CNN

Super-resolution is one of the famous vision applications in that CNNs have shown great promise. We applied the TPC-NAS method to search for the CNN architecture that effectively accomplishes super-resolution. However, super-resolution CNNs involve skip connections that are quite different from the CNN architectures for image classification. Note that the TPC score does not consider skip connections. Hence in our search for super-resolution CNN, we took the block of skip connection and concatenation in RFDN (Liu et al., 2020) as units and searched for different numbers of blocks, kernel size, channel size, etc. We used DIV2K with 800 training images as the training dataset. Furthermore, the batch size is 16, the image patch size is 64, and data augmentation, including random rotation and flipping, was applied. We adopted the following datasets for testing: SET5, SET14, B100, and Urban100.

Figure. 6 shows the super-resolution performance results of several models, including those found by TPC-NAS. In the experiment, we conducted the search under three hardware constraints: 300K, 500K, and 700K. In these scenarios, we discovered CNNs achieving 32.09dB, 32.31dB, and 32.44dB PSNR for the Urban100 dataset, respectively. Compared to other manually-designed models such as RFDN and CARN, the TPC-NAS models with comparable size outperform them by at least 0.2dB in PSNR. The TPC-NAS models also achieve significantly higher PSNR and SSIM than other NAS architectures with comparable computational costs (FLOPs), such as FALSR and ESRN. Finally, TPC-NAS takes only 3 CPU minutes, which is at least 3000 times faster than other NAS algorithms.

## 5 Discussion

### 5.1 Training Dataset

Since most zero-shot NAS methods do not require training data when searching, the dataset's information does not contribute to the model search process. In all likelihood, situations that the zero-shot NAS methods find the same architecture for quite different datasets/applications may occur. Such cases happen because zero-shot NAS methods seek models based on their expressivity, not their accuracy. As a result, we expect the TPC-NAS discovered models to be amenable to various ap-

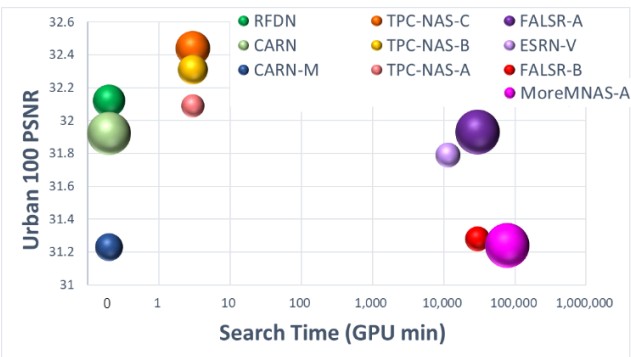

Figure 6: Super-resolution results for different NSA algorithms or manually-designed models. The x-coordinate is the search time on GPU (except TPC-NAS cases), the y-coordinate is the PSNR (in dB) of Urban 100 test datasets, and the bubble size represents the model's FLOPs. Models designed by humans have no search time. The complete comparison table is in Appendix A.4.

plications. However, we believe that utilizing the information in the dataset may still provide us with a better search outcome. It is likely that using TPC-NAS for coarse search and then other NAS algorithms, such as DARTS and one-shot algorithms, for fine and final search can achieve improved results.

## 5.2 MODEL QUANTIZATION

Model quantization is another popular topic in NN research. It can help reduce hardware computational and storage resources. As a result, the overall training and test time on hardware devices can be reduced. Furthermore, mixed-precision quantization (Sun et al., 2022; Yang et al., 2021; Yang & Jin, 2021; Uhlich et al., 2020) adopts different quantization precisions to different layers in the model, which compresses the model size to a larger degree. Along this line, we are studying zero-shot mixed-precision predictors that can be integrated into the proposed TPC-NAS solution. We hope to optimize the architecture and its quantization precision in a matter of minutes. Finally, current hardware constraints on FLOPs and the number of parameters can be extended to include weight and activation precisions, resulting in a more holistic NAS solution to finding low-complexity mixed-precision NNs.

## 5.3 HARDWARE-AWARE NAS

The hardware performance predictor can be used to predict the latency or power of a model on a specific hardware platform. Many NAS algorithms use MLP models (Lee et al., 2021; Nair et al., 2022) or lookup tables (Cai et al., 2019) to construct hardware performance predictors. However, an accurate predictor requires many resources and time, which may become the TPC-based NAS's bottleneck. In this case, the search time may increase from only five minutes to several hours. Toward this end, building a very efficient hardware predictor that estimates the power and latency of a task on a certain hardware platform would be essential for future work.

## 6 CONCLUSION

We propose a simple and effective TPC-NAS algorithm that does not require any architecture forward or backward propagation, allowing us to reduce the overall search time to less than five minutes on CPU. Furthermore, because the TPC score is based on simple model structural parameters, we can complete the entire NAS procedure on CPU or edge devices. Finally, to verify the proposed NAS solution, we apply it to three sophisticated vision applications: image classification, object detection, and super-resolution. In all three experiments, models found by TPC-NAS outperform those generated by almost all other NAS methods and those designed by experts, except for a couple of cases.

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

# A APPENDIX

## A.1 SEARCH SPACE

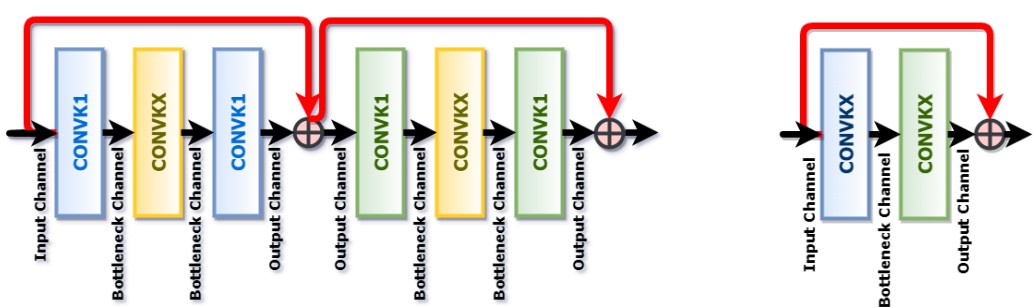

Figure 7: Building blocks CONVK1KXK1 (left) and CONVKXKX (right). The FLOP of the CONVKX layer is proportional to the layer's input and output channels. With the bottleneck channel, the computation cost of CONVKX can be reduced significantly.

This Appendix will present the TPC-NAS search space used in each of the three applications.

We used CONVK1KXK1 and CONVKXKX, shown in Figure 7, as the basic blocks of the architectures for the image classification problem. CONVK1KXK1 consists of two convolutional layers with a kernel size of 1 as the front and back end. It also includes a bottleneck channel that is usually much smaller than the input and output channels as the intermediate convolutional layer. This structure can significantly reduce the required computation. In the same vein, we also adopted a bottleneck channel in the middle of CONVKXKX structure. For the search space, we build the target architectures for image classification by stacking a number of these two basic blocks. In addition, a final FC layer is concatenated to generate scores for different image classes.

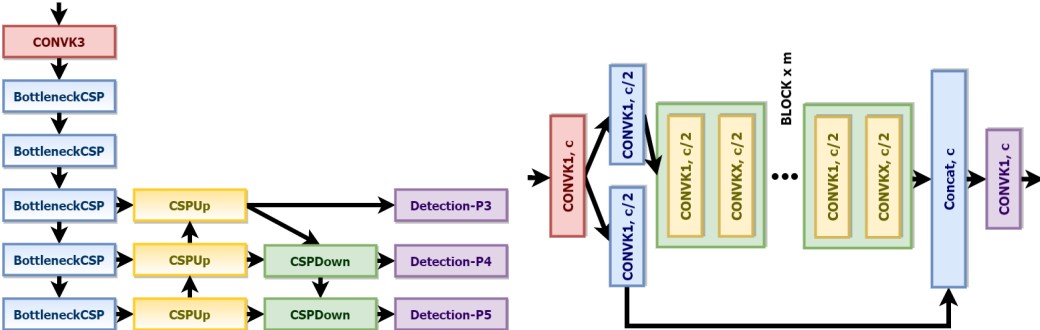

Figure 8: Overall architecture (left) and BottleneckCSP structure (right) of Yolov4-p5. We set all the connections the same as Yolov4-p5. However, the number of blocks and the block's parameters, such as kernel size, channel are changed in TPC-NAS.

To find good CNN solutions to the object detection problem, we selected the YOLOv4-p5, which has excellent performance, albeit relatively high computation cost, as the initial architecture. The overall architecture of YOLOv4-p5 is shown on the left side of Figure A.1. The backbone consists of several BottleneckCSP modules, which the mutation function focuses on. The right side of Figure A.1 is the architecture of the bottleneckCSP module, where $c$ represents the channel size. By changing the depth, the number of channels, kernel size, and other characteristics of the BottleneckCSP modules in the YOLOv4-p5 architecture, we could generate more candidates for the search space. Finally, note that the block type and the connection method between the blocks remain the same as the original YOLOv4-p5.

In the experiment of searching for enhanced super-resolution CNNs, we picked the RFDN architecture, shown in Figure 9, as the baseline structure. On the left of Figure 9 is the entire RFDN

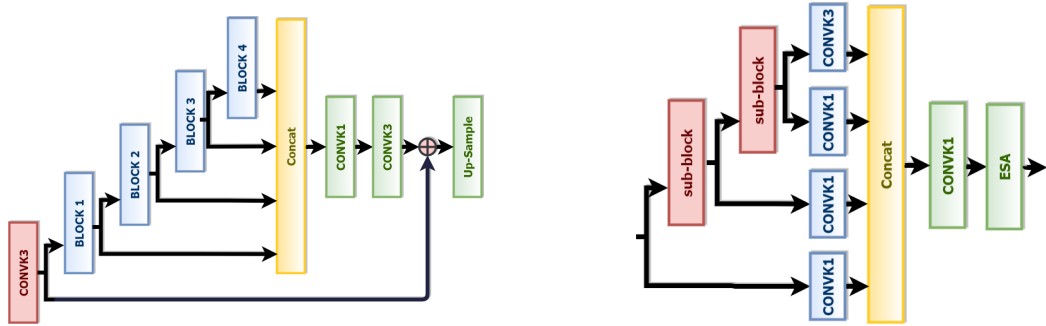

Figure 9: Overall architecture (left) and block structure (right) of RFDN. We set all the connections the same as RFDN. However, the number of blocks and the block's parameters, such as depth, width, and bottleneck channel, are changed in TPC-NAS.

architecture, consisting of multiple basic blocks and a wide concatenation layer. The basic block on the right of Figure 9 consists of multiple sub-blocks and a concatenation layer. To construct the search space, we first varied the number of blocks in the entire architecture and the number of sub-blocks in each block. Then, within each sub-block, we changed its channel number, kernel size, and other characteristics.

## A.2 IMAGE CLASSIFICATIONT EXPERIMENTAL RESULTS

Table. 2 list the search time and achieved classification accuracy for the architectures discovered by several NAS methods. TPC-NAS can complete the search process within two minutes of CPU time. The architecture found by TPC-NAS is among the best solutions, beating a couple of well-known low-complexity CNNs.

Table 2: ImageNet accuracy and search time of different NAS algorithms

| Models | FLOPs (M) | Top-1 Accuracy (%) | Search Time (GPU min) |
|---|---|---|---|
| MobileNetV2 | 314 | 72.8 | – |
| ShuffleNetV2 | 299 | 72.6 | – |
| MnasNet-92 | 388 | 74.8 | 5.7 M |
| DARTS | 574 | 73.3 | 720 |
| PC-DARTS | 586 | 74.9 | 144 |
| SPOS | 323 | 74.4 | 17 K |
| Single Path NAS | 365 | 75.0 | 1.8 K |
| ProxylessNAS | 465 | 75.1 | 12.0 K |
| OFA w/ PS | 230 | 76.0 | 250 K |
| AtomNAS-B | 326 | 75.5 | 29.5 K |
| FBNet-B | 295 | 74.1 | 13.0 K |
| TE-NAS | – | 75.5 | 244.8 |
| Zen-NAS | 410 | 78.0 | 720 |
| **TPC-NAS** | **355** | **76.4** | **2 (CPU)** |

## A.3 OBJECT DETECTION EXPERIMENTAL RESULTS

Table 3 lists the experiment results of object detection models by a few NAS methods. TPC-NAS completes its search in 0.002 GPU days, more than 1000 times faster than other NAS algorithms. Furthermore, the model found, TPC-YOLO-s and TPC-YOLO-m, achieve 43.4 and 45.4 mAP under 20G and 40G FLOPs, respectively, higher than similarly sized architectures. This demonstrates that TPC-NAS can further improve previously-designed architecture while maintaining all of the architecture blocks and changing only some parameters, such as depth, the number of channels, kernel size, etc.

Table 3: The object detection precision and search time results of TPC-NAS from YOLOv4-p5 and other object detection architectures. For manually-designed models, no search time is listed.

| Model | Backbone | FPN | #Param (M) | FLOPs (G) | Resolution | mAP (%) | AP50 (%) | Search Cost (GPU days) |
|---|---|---|---|---|---|---|---|---|
| YOLOv5-s | YOLOv5 | PAN | 7.3 | 17.1 | 640 | 36.9 | 56.0 | – |
| EAutoDet-s | Searched | Searched | 9.1 | 24.9 | 640 | 40.1 | 58.7 | 1.4 |
| EfficientDet-D0 | Effieicent-B0 | BiFPN | 3.9 | 2.5 | 512 | 33.8 | 52.2 | – |
| YOLOv4 | CD-53 | PAN | – | 60.1 | 416 | 41.2 | 62.8 | – |
| **TPC-YOLO-s** | **Searched** | **PAN** | **19.7** | **20.1** | **640** | **43.4** | **61.3** | **0.002** |
| YOLOv5-m | YOLOv5 | PAN | 21.4 | 51.4 | 640 | 43.9 | 62.5 | – |
| YOLOv4-p5 | CSP-P5 | PAN | 70.8 | 163.9 | 896 | 51.8 | 70.3 | |
| YOLOv4 | CD-53 | PAN | – | 91.1 | 512 | 43.0 | 64.9 | – |
| EAutoDet-m | Searched | Searched | 28.1 | 60.8 | 640 | 45.2 | 63.5 | 1.4 |
| NAS-FPN | Resnet-50 | Searched | 60.3 | 281.3 | 640 | 39.9 | – | 333 |
| SM-NAS:E2 | Searched | – | – | – | 800*600 | 40.0 | 58.2 | 80 |
| SM-NAS:E3 | Searched | – | – | – | 800*600 | 42.8 | 61.2 | 80 |
| EfficientDet-D1 | Effieicent-B1 | BiFPN | 6.6 | 6.1 | 640 | 39.6 | 58.6 | – |
| **TPC-YOLO-m** | **Searched** | **PAN** | **37.0** | **40.2** | **640** | **45.4** | **64.0** | **0.002** |

## A.4 SUPER-RESOLUTION EXPERIMENTAL RESULTS

Table 4 shows that TPC-NAS can also generate high-performance RFDN-based models for the super-resolution application. The TPC-NAS-A/B/C models beat other designer models and NAS-discovered models in terms of computational complexity. In addition, the TPC-NAS search time is much shorter than other NAS algorithms.

Table 4: Average PSNR/SSIM on datasets SET5, SET14, B100, and Urban100 of CNN models for super-resolution. TPC-NAS-A/B/C are searched under 300K, 500K, and 700K parameter size constraints, respectively.

| | FLOPs | Param | Urban100 | B100 | SET14 | SET5 | Search time (GPU hours) |
|---|---|---|---|---|---|---|---|
| **CARN** | 222.8 G | 1592 K | 31.92/0.926 | 32.09/0.898 | 33.52/0.917 | 37.76/0.959 | – |
| **CARN-M** | 91.2 G | 412 K | 31.23/0.919 | 31.92/0.896 | 33.26/0.914 | 37.53/0.958 | – |
| **RFDN** | 123 G | 534 K | 32.12/0.928 | 32.16/0.899 | 33.68/0.918 | 38.05/0.961 | – |
| **ESRN-V** | 73.4 G | 324 K | 31.79/0.925 | 32.10/0.899 | 33.42/0.916 | 37.85/0.960 | 192 |
| **MoreMNAS-A** | 238.6 G | 1039 K | 31.24/0.919 | 31.95/0.896 | 33.23/0.914 | 37.63/0.958 | 1300 |
| **FALSR-A** | 234.7 G | 1021 K | 31.93/0.927 | 32.12/0.897 | 33.55/0.917 | 37.82/0.960 | 500 |
| **FALSR-B** | 74.7 G | 326 K | 31.28/0.919 | 31.97/0.897 | 33.29/0.914 | 37.61/0.959 | 500 |
| **FALSR-C** | 93.7 G | 408 K | 31.24/0.918 | 31.96/0.897 | 33.26/0.914 | 37.66/0.959 | 500 |
| **TPC-NAS-A** | **62 G** | **298 K** | **32.09/0.928** | **32.15/0.900** | **33.57/0.918** | **37.91/0.961** | **0.05** |
| **TPC-NAS-B** | **104 G** | **500 K** | **32.31/0.930** | **32.20/0.900** | **33.69/0.918** | **38.04/0.961** | **0.05** |
| **TPC-NAS-C** | **140 G** | **690 K** | **32.44/0.931** | **32.23/0.900** | **33.73/0.918** | **38.05/0.961** | **0.05** |

## A.5 ABLATION STUDIES

The first ablation study we conducted was to vary the hardware constraints during Resnet-based search and training with the CIFAR10 and CIFAR100 datasets. Tables 5 and 6 list accuracies of CNN models found by the TPC-NAS method. As expected, when we allowed higher FLOPs or Params constraints, architectures with higher TPC scores were found, and they achieved higher accuracies.

Table 5: FLOPs and Accuracy

| FLOPs | TPC score | CIFAR10 | CIFAR100 |
|-------|-----------|---------|----------|
| 200M  | 480       | 94.17   | 76.86    |
| 100M  | 413       | 92.62   | 74.77    |
| 50M   | 379       | 91.88   | 71.93    |

Table 6: Params and Accuracy

| Params | TPC score | CIFAR10 | CIFAR100 |
|--------|-----------|---------|----------|
| 2M     | 579       | 95.33   | 77.68    |
| 1M     | 489       | 94.27   | 75.04    |
| 0.5M   | 403       | 93.21   | 70.89    |

Next, we varied the maximal depths and again trained on the CIAFR10 and CIFAR100 datasets. The results are shown in Table 7. We find that the TPC score grows as the depth increases. However, the accuracy will stagger and not increase, which we think is due to the difficulty in training deeper architectures. Toward this end, our TPC-NAS search space has a depth limit on the architectures.

Table 7: Maximal Depth and Accuracy

| Depth | TPC score | CIFAR10 | CIFAR100 |
|-------|-----------|---------|----------|
| 12    | 369       | 93.95   | 76.43    |
| 13    | 401       | 94.42   | 77.09    |
| 14    | 416       | 94.16   | 76.15    |
| 15    | 468       | 93.82   | 76.02    |

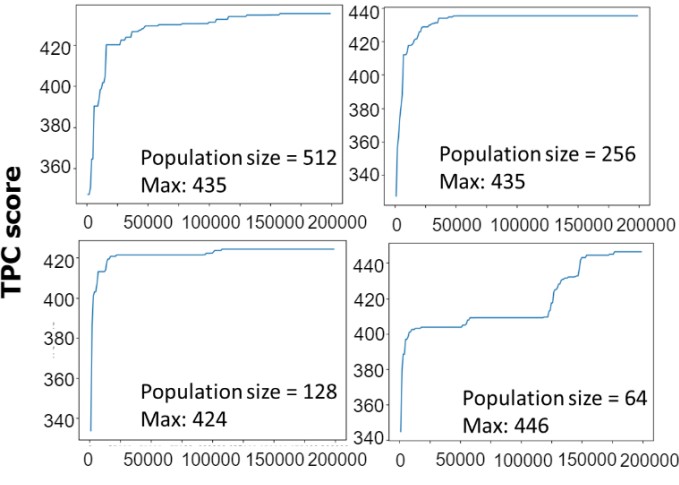

Figure 10: TPC outcomes of the TPC-NAS search with different population sizes under the conditions that the iteration number is 200000, FLOPs budget is 200M and the maximum number of layers is 14.

We modified the population size for the evolutionary search with 200000 iterations, up to 200M FLOPs, and 14 layers. As shown in Figure 10, a larger population size does not necessarily lead to a better architecture. We think that with 200000 architectures searched, the population size has no bearing on the final outcome. This again evidences the TPC score's power in quickly evaluating the expressivity of so many architectures.

The adopted search space is of course crucial to the success of any NAS solution. Thanks to the TPC score, a search space can be explored very quickly. Therefore, if the search space is not adequate, the TPC score quickly reveals that and proper modification can be made.

For the initial structure, we adopted a small structure in the search space. Then we allowed the search to iterate for a large number of epochs since the TPC score computation takes very little time. Therefore, no matter the initial structure, the search procedure always converges at architectures with high TPC scores.

### A.6   COMPARISON OF EXISTING MODELS

We identified several well-known architectures for image classification using the Imagenet database: Resent-18/50, MobileNetV2, and ShuffleNetV2. Then we calculated their TPC scores and listed the scores with their top-1 accuracy performance on the ImageNet database in Table 8. Once again, the TPC score successfully predicts accuracy performance with high correlation even on the architectures with different structures.

Table 8: Comparison of Existing Models

| Model | TPC score | ImageNet |
|-------|-----------|----------|
| Resnet-18 | 140 | 69.76 |
| MobileNet_V2 | 214 | 71.88 |
| ShuffleNet_V2_1.5 | 235 | 73.00 |
| Resnet-50 | 340 | 76.13 |

