# OpenReview forum: "TPC-NAS: Sub-Five-Minute Neural Architecture Search for Image Classification, Object-Detection, and Super-Resolution"
_ICLR.cc/2023/Conference — Submitted to ICLR 2023_

### Official Review · Reviewer_F9pB · 2022-10-22

**Confidence:** 4
**Correctness:** 4
**Technical Novelty And Significance:** 3
**Empirical Novelty And Significance:** 3
**Recommendation:** 5

**Clarity, Quality, Novelty And Reproducibility:**

The paper is written clearly and is novel. The authors provided the code. So it is reproducible.

**Strength And Weaknesses:**

**Strengths**
1. It is almost instantaneous to compute the metric for a given architecture
2. The experiments demonstrated that TPC-NAS is able to find well performing networks in ResNet search space to solve image classification and object detection.

**Weakness**
1. In Figure 3, the Kendall tau is computed on only 20 architectures. We need at least 100 architectures for the correlation to be significant.
2. NASWOT and TE-NAS are computed on NasBench 101 and 201. So please demonstrate how effective your metric is on those 2 search spaces for at least 2 datasets. Also, please don't restrict the number of parameters
3. As mentioned in Section 5.1, when the dataset is not taken into consideration while computing the proxy metric, then the architectures would be ranked the same on all the datasets. The authors suggest that TPC score could be used to eliminate bad candidate architectures early on. But for a good proxy metric, one must take into consideration the dataset characteristics. As mentioned in 2, please compute Kendall Tau for two datasets on the same search space and compare them against the other baselines.
4. TPC Score only takes CONV layers into consideration. What about the impact of dropout, pool or skip connections? Initialization also plays a crucial role in the final performance of the network.

**Summary Of The Paper:**

This paper proposes a training free algorithm for Neural Architecture Search (NAS). The proxy metric (TPC) used is the number of paths from the input to the output layer. To generalize it for very large networks, the network can be viewed as a graph and one could compute the outdegree of a node. This score is then used to rank all the architectures. TPC-Score was used a proxy instead of validation accuracy in  an evolutionary algorithm and neural architecture search was performed.  For image classification, they used ResNet search space where they constrained the number of model parameters to 1M. In addition to this, TPC-NAS was used to search for object detection architecture in the YOLOv4 search space. This metrics is very faster compared to other training-free proxies.

**Summary Of The Review:**

This paper devised a training-free NAS algorithm, where the metric is the number of paths from the input to the output layer. But its performance needs to be tested on NASBench-101 and 201 to be fair to other baselines. It is also important to understand if it adapts its score to a given dataset as they do not consider any dataset characteristics.

---

> ### Author Response · Authors · 2022-11-18
> **Responses to Official Reviewer F9pB**
>
> We would like to thank the reviewer for the useful feedback and valuable comments that have helped improve the quality of this paper. In addition to addressing all the reviewer’s comments, we have improved the presentation of the paper and made our results more convincing. Below, we list the point-by-point responses to all the reviewer’s comments. In addition, all revisions in the manuscript are highlighted in red.
>
> 1.	It is almost instantaneous to compute the metric for a given architecture. The experiments demonstrated that TPC-NAS is able to find well performing networks in ResNet search space to solve image classification and object detection.
> Response:
> Thank you very much for your comments.
>
> 2.	In Figure 3, the Kendall tau is computed on only 20 architectures. We need at least 100 architectures for the correlation to be significant.
> Response:
> Thank you very much for your comment. It would be nice to have 100 architectures for computing the Kendall’s rank correlation coefficient. However, we want to compare the TPC score with the score used in the ZenNAS paper, and since only 20 architectures were used in that paper, so for a fair comparison we still adopted 20 architectures in plotting Fig. 3.
>
> 3.	NASWOT and TE-NAS are computed on NasBench 101 and 201. So please demonstrate how effective your metric is on those 2 search spaces for at least 2 datasets. Also, please don't restrict the number of parameters
> Response:
> Thank you very much for your comment. We have also tried NAS-Bench-201 as the sample architectures for the search space to verify the validity of the TPC score. We selected the architectures with a prominent backbone, and keep only the architectures achieving higher than 80% accuracy in CIFAR10. The outcomes are depicted in https://pasteboard.co/5HoCbw9kN6ea.png, and the Kendall rank coefficient is 0.413. We find that there are many architectures with the same TPC score. This is because most networks in NAS-Bench-201 vary only in the block type, and do not consider changing the channel numbers or the architecture depth. However, channel number and architecture depth are the very core of the TPC score, which explains its impaired accuracy prediction ability for NAS-Bench-201.
>
>
> 4.	As mentioned in Section 5.1, when the dataset is not taken into consideration while computing the proxy metric, then the architectures would be ranked the same on all the datasets. The authors suggest that TPC score could be used to eliminate bad candidate architectures early on. But for a good proxy metric, one must take into consideration the dataset characteristics. As mentioned in 2, please compute Kendall Tau for two datasets on the same search space and compare them against the other baselines.
> Response:
> Thank you very much for your comment. In the limited timeframe, we have made the following attempts. We randomly chose several architectures from our search space and trained them with CIFAR10 and CIFAR100 datasets on these architectures. Their accuracies are plotted in https://pasteboard.co/jcZLDW11fF69.png. We can see from the Kendall rank coefficient between accuracies of the CIFAR10 and CIFAR100 datasets that an architecture with a relatively high TPC score can perform well in two datasets, implying that a highly-expressive architecture can perform well in different datasets.
>
> 5.	TPC score only takes CONV layers into consideration. What about the impact of dropout, pool or skip connections? Initialization also plays a crucial role in the final performance of the network.
> Response:
> Thank you very much for your comments. In our opinion, the dropout operation is more pertinent to training, and less so for architectural structure. Were we to consider the dropout operation, then we need to consider for all competing architectures. And we think that the comparison outcome may be the same as when we ignore the dropout operation. As such, we did not consider the dropout operation in the TPC score.
>
> For the max pool function, we did not change the position or the existence of the max pool function in the search spaces. Therefore, the maxpool function is also not included in the TPC score.
>
> We think that skip connections affect only the ease of training, but not the model's expressivity. The number of paths contributed by skip connections is significantly less that that of the backbone. Hence, ignoring skip connections when calculating the TPC score does not affect the TPC score’s validity in quantifying an architecture’s expressivity.
>
> For the initial structure, we adopted a small structure in the search space. Then we allowed the search to iterate for a large number of epochs since the TPC score computation take very little time. Therefore, no matter the initial structure, the search procedure always converges at architectures with high TPC scores.

---

> > ### Comment · Reviewer_F9pB · 2022-12-13
> > **Thank you for your response**
> >
> > I thank the authors for their response. I do understand that ZenNAS used only 20 architectures to compute the correlation, but that does not make it right.This method works well when the architectures vary in their depth and number of channels etc. As is evident from Kendall Tau on NAS-Bench 201, this method is not adequate to rank networks if the backbone of the architecture is similar but the operations vary. So I retain my score.

---

### Official Review · Reviewer_CdaL · 2022-10-25

**Confidence:** 4
**Clarity, Quality, Novelty And Reproducibility:** An anonymous GitHub link is included …
**Correctness:** 3
**Technical Novelty And Significance:** 3
**Empirical Novelty And Significance:** 3
**Recommendation:** 5

**Strength And Weaknesses:**

Strength

- proposes the total path count (TPC) score as an efficient accuracy predictor for neural architecture search

- outperforms the most relevant hand-crafted and NAS-discovered architectures on three applications

Weaknesses

- Clarity needs to be improved. Some important sections/functions are not well explained. For example, there is not a detailed explanation regarding the MUTATE function. It would be better to include an explanation to make the paper to be self-contained.

- Experiments are lacking. There are no ablations on the hyperparameters, like Search Space S, hardware constraints K, maximal depth D, number of iterations M, population size N, and initial structure F0. The paper only compares the best results with the state-of-the-art approaches but does not explain the sensitivity to the hyperparameters. It is hard to justify the superior performance of the proposed approaches without giving the hyperparameters.


**Summary Of The Paper:**

This paper proposes the total path count (TPC) score as an efficient accuracy predictor for neural architecture search. Particularly, the TPC score is defined as the number of paths of the ”vanilla” convolutional neural network, which is very simple to come by. In experiments, the proposed approach was evaluated on three vision tasks: image classification, object detection, and super-resolution.

**Summary Of The Review:**

This paper proposes the total path count (TPC) score as an efficient accuracy predictor for neural architecture search and outperforms the most relevant hand-crafted and NAS-discovered architectures on three vision applications. But clarity needs to be improved and experiments are lacking.

---

> ### Author Response · Authors · 2022-11-18
> **Responses to Official Reviewer CdaL**
>
> We would like to thank the reviewer for the useful feedback and valuable comments that have helped improve the quality of this paper. In addition to addressing all the reviewer’s comments, we have improved the presentation of the paper and made our results more convincing. Below, we list the point-by-point responses to all the reviewer’s comments. In addition, all revisions in the manuscript are highlighted in red.
>
> 1.	Clarity needs to be improved. Some important sections/functions are not well explained. For example, there is not a detailed explanation regarding the MUTATE function. It would be better to include an explanation to make the paper to be self-contained.
> Response:
> Thank you for the comment. In the revised manuscript, we have added an algorithm description of the MUTATE function used in the NAS procedure. Moreover, the first paragraph of Sec. 3.5 now reads
> ……. The initial model F0 is typically set as a small architecture in the search space. The MUTATE function creates new models from the search space according to Algorithm 3. A new model …..
>
> We have also checked the whole manuscript and improved the clarity of the presentation, such as the search spaces for the three TPC-NAS tasks are now described in more detail (Appendix A.1).
>
> 2.	Experiments are lacking. There are no ablations on the hyperparameters, like Search Space S, hardware constraints K, maximal depth D, number of iterations M, population size N, and initial structure F0. The paper only compares the best results with the state-of-the-art approaches but does not explain the sensitivity to the hyperparameters. It is hard to justify the superior performance of the proposed approaches without giving the hyperparameters.
> Response:
> Thank you for your suggestion. We have added a new subsection in the Appendix to discuss ablation studies on search space, FLOPS/number of parameters, maximal depth, number of iterations, population size, and initial structure. Please refer to Appendix A.5 of the revised manuscript.

---

### Official Review · Reviewer_3RDB · 2022-11-02

**Confidence:** 5
**Clarity, Quality, Novelty And Reproducibility:** The manuscript is of moderate quality…
**Correctness:** 4
**Technical Novelty And Significance:** 1
**Empirical Novelty And Significance:** 2
**Recommendation:** 5

**Details Of Ethics Concerns:**

N/A.

**Strength And Weaknesses:**

<Strength>

•	The paper is well-organized and well-written.

•	The experimental results supports the claims made in the manuscripts.

•	The submitted content is related to the application of fundamental tasks in computer vision, which is highly relevant to the ICLR audience.

<Weakness>

•	The novelties of the proposed algorithm is limited. Computing the TPC score is similar to study the effective receptive field of CNN models. It is a fact that large receptive field helps models’ expressivity because of long-range dependencies between image pixels.

•	The impact of the proposed approaches could be limited. Because it is applied to CNN models only and within a relative small search space.

•	Discussion or ablation studies are not sufficient (see below).

<Comments>

1.	How to extend the TPC score to the popular neural networks like various vision transformers or models with self-attention layers?

2.	Does the training recipe matter for the NAS outcome? Does different training recipes or with different random seeds result in different Kendall rank correlation coefficients?

3.	The manuscript claimed that the TPC score can be computed within a few microseconds. Please further clarify the device for such computation (e.g., what type of CPU or GPU)?

4.	Does the TPC score correlate with the accuracy in a higher order? For instance, does large TPC score gap mean larger accuracy gap?

5.	What are the (approximated) Kendall rank correlation coefficients for object detection and super-resolution?

6.	It is important to get a high ranking correlation for the model candidates with top performance because they matters more for final architecture selection. It would be better to compute the Kendall ranking coefficients for the models with top performance (e.g., top 5~10% accuracies).

7.	What are the typical cases for good models with low TPC scores? What are the reasons causing the low scores for the models?

8.	Can the TPC score be used for comparison of different CNN models (e.g., ResNet versus Inception network)?

9.	Typo: Page 4, “RELU” => “ReLU”.


**Summary Of The Paper:**

The manuscript proposed an efficient neural architecture search (NAS) algorithm for computer vision tasks including 2D image classification, object detection, and super-resolution. The proposed algorithm introduces the total path count (TPC) score as an accuracy predictor. The TPC score is computed based on the shape of convolution kernels to indicate models’ expressivity. And its computation can be efficiently achieved within few minutes. Moreover, the experimental results validate the proposed algorithm with several datasets of different computer vision tasks. All experiments use convolutional neural network (CNN) based search space.

**Summary Of The Review:**

My concerns about the manuscript are about its novelties and practical impacts. The novelties of the proposed algorithm is limited because the TPC score concept is very similar to the model’s receptive field. And the proposed algorithm may not be practical as it is limited for CNN models only and applied within a small search space.

---

> ### Author Response · Authors · 2022-11-18
> **Responses to Official Reviewer 3RDB**
>
> We would like to thank the reviewer for the helpful feedback and many valuable comments that have helped improve the quality of this paper. In addition to addressing all the reviewer’s comments, we have improved the presentation of the paper and made our results more convincing. Below, we list the point-by-point responses to all the reviewer’s comments. In addition, all revisions in the manuscript are highlighted in red.
>
> 1.	The paper is well-organized and well-written. The experimental results support the claims made in the manuscripts.
> Response:
> Thank you very much for your comments.
>
> 2.	The novelties of the proposed algorithm is limited. Computing the TPC score is similar to study the effective receptive field of CNN models. It is a fact that large receptive field helps models’ expressivity because of long-range dependencies between image pixels.
> Response:
> Thank you very much for your comment. In the following, we would like to explain the difference between the receptive field and the total path count (TPC).
>  According to Wikipedia [1] “In a neural network context, the receptive field is defined as the size of the region in the input that produces the feature. Basically, it is a measure of association of an output feature (of any layer) to the input region (patch).”
> As an example, we depict two neural networks in the figure from https://pasteboard.co/qmrbcWN67FSj.png. Both NNs have the same receptive field, which includes all three input neurons. On the other hand, the TPC score of NN on the left is 4*2 =8, while the TPC score of the NN on the right is 2*2=4. It is evident that the NN with a higher TPC score also has better expressivity, thus showing the effectiveness of the proposed TPC score.
>
> 3.	 The impact of the proposed approaches could be limited. Because it is applied to CNN models only and within a relative small search space.
> Response:
> Thank you for your comment. We have actually begun experimenting with applying our TPC-NAS to other NN categories, such as RNN and GAN. We are hopeful that NAS for GAN can also benefit from the TPC score since GANs consist mainly of convolutional layers. As for the search space size used in the experiments, there is no upper limit on structural parameters. With possibly an infinite number of channels and an unlimited number of layers and hardware, the search space is far from small. Moreover, we have reached 400,000 epochs using evolutionary search, indicating that the number of architectures we have generated in this search space is quite large.
>
> 4.	Discussion or ablation studies are not sufficient (see below).
> Response:
> Thank you for your suggestion. We have added two ablation studies in the Appendix.
>
> 5.	How to extend the TPC score to the popular neural networks like various vision transformers or models with self-attention layers?
> Response:
> Thank you for your comment. The transformer or architecture with a self-attention layer is undeniably an important part of the AI field. Still, because its architecture is much more complicated than the previous CNN, we find it difficult to express the functions of different layers with simple mathematical expressions. However, most transformer architectures still retain a primary "feed-forward" backbone. We can concentrate the NAS procedure on the backbone and apply the TPC score to find the backbone architecture with the highest expressivity.
>
> 6.	Does the training recipe matter for the NAS outcome? Does different training recipes or with different random seeds result in different Kendall rank correlation coefficients?
> Response:
> Thank you for your comment. The training recipe can indeed affect the trained architecture's accuracy performance. However, when calculating the Kendall rank coefficient in Fig. 3 and the comparison in Table 1, we adopted the search space and training mechanism as in the Zen-NAS work (Lin et al., 2021). We believe that with enough number of architectures, the Kendall rank coefficient should stay approximately the same with different random seeds.
>
> 7.	The manuscript claimed that the TPC score can be computed within a few microseconds. Please further clarify the device for such computation (e.g., what type of CPU or GPU)?
> Response:
> We only used a CPU when implementing the TPC-NAS procedure. The CPU used is Intel(R) Xeon(R) CPU E5-2620 v3 @ 2.40GHz with 32GB DRAM.
>
> 8.	Does the TPC score correlate with the accuracy in a higher order? For instance, does large TPC score gap mean larger accuracy gap?
> Response:
> Thank you for your comment. Figure 3 shows that, as the TPC Score increases, the accuracy increases rapidly at first but then gradually converges to a high value. At this point, even if the TPC score continues to rise, the accuracy will not improve significantly. We conclude that the lower the TPC score, the larger the corresponding accuracy gap. On the other hand, the accuracy tends to saturate to a high value, with many architectures achieving such performance.

---

> > ### Author Response · Authors · 2022-11-18
> > **Responses to Official Reviewer 3RDB (continued)**
> >
> > 9.	What are the (approximated) Kendall rank correlation coefficients for object detection and super-resolution?
> > Response:
> > Thank you for your comment. Currently, our computing resources prevent us from training dozens of different architectures in the search space for object detection or super-resolution. It may take months to do so. In the limited timeframe available to us, we thus did not carry out this task. However, we are confident that the Kendall rank correlation coefficients will still be high, judging from the high-performance architectures we have found for the two applications based on the TPC score, as shown in Figs. 5 and 6.
> >
> > 10.	It is important to get a high ranking correlation for the model candidates with top performance because they matters more for final architecture selection. It would be better to compute the Kendall ranking coefficients for the models with top performance (e.g., top 5~10% accuracies).
> > Response:
> > Thank you for your comment. As the reply to the previous comment on high-order correlation stated, top-scoring architectures tend to have saturating accuracy performance, as shown in Fig. 3.  Therefore, the top 5%-10% of the best architectures may have an accuracy difference of less than 1%. Practically, even with the wrong ranking order in these top architectures, the final outcome of the search will still present high-accuracy architectures.
> >
> > 11.	What are the typical cases for good models with low TPC scores? What are the reasons causing the low scores for the models?
> > Response:
> > As shown in Figure 3, there were no apparent counter-examples where good models have low TPC scores. However, if we do not put a limit to the architecture depth, we can encounter very deep architectures with very high TPC scores but poor accuracy after training. As such, practically, there is a limit to the architecture depth during TPC-NAS. This does not imply that the deep architecture has poor expressivity. Instead, it could be that the architecture is too deep for us to train or that we do not have enough data to allow the model to achieve its full expressivity potential.
> >
> > 12.	Can the TPC score be used for comparison of different CNN models (e.g., ResNet versus Inception network)?
> > Response:
> > Thank you for your comment. We have a simple example in our paper. We compare the TPC score of separable convolution to that of the conventional convolution. The scores are found to be the same, which is in line with the consensus that the separable CONV layer achieves similar performance to the conventional CONV layer.
> > In addition, we have computed the TPC scores for Resenet-18/50, MobileNetV2, and ShuffleNetV2. These scores and the corresponding Imagenet classification accuracies are listed in Table 8 of the revised manuscript. Clearly, these data again verify the high correlation between the TPC scores and the accuracies achieved by rather different CNN models.
> > Finally, we have included the above in Appendix A.6.
> >
> > 13.	Typo: Page 4, “RELU” => “ReLU”.
> > Response:
> > Thank you for the reminder. We have corrected this typo.
> >
> > Reference:
> > [1]. “Receptive Field - Wikipedia.” Receptive Field - Wikipedia, en.wikipedia.org/wiki/Receptive_field. Accessed 9 Nov. 2022.

---

### Official Review · Reviewer_3zBH · 2022-11-02

**Confidence:** 4
**Correctness:** 3
**Technical Novelty And Significance:** 3
**Empirical Novelty And Significance:** 2
**Recommendation:** 5

**Clarity, Quality, Novelty And Reproducibility:**

The paper is overall well written and easy to follow. But, some experiments need further clarifications, as described above. The TPC scoring method, and application to object detection and super-resolution seem to be the main contributions of this work. Code is also provided for reproducibility.

**Strength And Weaknesses:**

This paper introduces the TPC as an extremely efficient architecture performance predictor that depends only on the architecture topology. This can significantly speed up NAS algorithms when combined with different black box search strategies.

However, the Kendall tau analysis in section 3.4 is very limited, and the search space is not clearly described. Extending the experiments would be useful, in order to have a better assessment of how TPC performs in ranking architectures. One can for example sample architectures from CNN NAS search spaces such as NAS-Bench-201 whose architecture test accuracies can be queried as well.

Furthermore, in the experiments section it is not clearly stated whether the search spaces are the same for all methods in table.1. In general, in section 4, further experiments could be useful to clarify whether it is the TPC scoring method, the evolutionary search strategy, or the chosen search space which plays the most important role in achieving high performance on different tasks.

(Also, the mutation operation in algorithm 2 is not defined, and in the main text there is no reference to the search space defined in the appendix.)

**Summary Of The Paper:**

In this work the Authors introduce the total path count (TPC) of a network as a zero-cost measure of network performance that can be used to rank CNN architectures very efficiently.

They further combine the TPC as a performance predictor with an evolutionary algorithm as a search strategy to introduce a new architecture search method (TPC-NAS).

TPC-NAS is then used to find high-performing architectures with hardware constraints on image classification, object detection and super-resolution tasks.

**Summary Of The Review:**

The efficient architecture performance predictor introduced in this paper is interesting and can in principle be very useful for neural architecture search if it is proven to be effective. However, experiments in section 3.4 are not extensive enough to support this fact. Moreover, some confusions need to be addressed in the experiments section. I would therefore rate this paper as marginally below acceptance threshold.

---

> ### Author Response · Authors · 2022-11-18
> **Responses to Official Reviewer 3zBH**
>
> We would like to thank the reviewer for the useful feedback and valuable comments that have helped improve the quality of this paper. In addition to addressing all the reviewer’s comments, we have improved the presentation of the paper and made our results more convincing. Below, we list the point-by-point responses to all the reviewer’s comments. In addition, all revisions in the manuscript are highlighted in red.
>
> 1.	This paper introduces the TPC as an extremely efficient architecture performance predictor that depends only on the architecture topology. This can significantly speed up NAS algorithms when combined with different black box search strategies.
> Response:
> Thank you very much for your comment. We also think that the TPC score can benefit a lot of other NAS researchers due to its computational efficiency.
>
> 2.	However, the Kendall tau analysis in section 3.4 is very limited, and the search space is not clearly described. Extending the experiments would be useful, in order to have a better assessment of how TPC performs in ranking architectures. One can for example sample architectures from CNN NAS search spaces such as NAS-Bench-201 whose architecture test accuracies can be queried as well.
> Response:
> Thank you very much for your comments.
> In the experiments for Fig. 3, everything is based on the Zen-NAS settings except for the proxy metric (TPC score), so the search space is the same as in the Zen-NAS paper. We have explained this in the revised manuscript. Fig. 3 caption now reads:
>
> TPC score v.s. CIFAR100 accuracy. For this evaluation, we randomly sampled 20 structures from the Resnet search space (Appendix A.1), which are the same as those used in Zen-NAS work (Lin et al., 2021).
>
> We have also tried NAS-Bench-201 as the sample architectures for the search space to verify the validity of the TPC score. We selected the architectures with a prominent backbone, and kept only the architectures achieving higher than 80% accuracy in CIFAR10. The outcomes are depicted in https://pasteboard.co/5HoCbw9kN6ea.png, and the Kendall rank coefficient is 0.413. We find that there are many architectures with the same TPC score. This is because most networks in NAS-Bench-201 vary only in the block type, and do not consider changing the channel numbers or the architecture depth. However, channel number and architecture depth are the very core of the TPC score, which explains its impaired accuracy prediction ability for NAS-Bench-201.
>
> 3.	Furthermore, in the experiments section it is not clearly stated whether the search spaces are the same for all methods in table.1. In general, in section 4, further experiments could be useful to clarify whether it is the TPC scoring method, the evolutionary search strategy, or the chosen search space which plays the most important role in achieving high performance on different tasks.
> Response:
> Thank you very much for your comments. We have noted in the main text the search space used in Table 1. Now the first paragraph of Sec. 4.1 is revised as
>
> …epochs of training. In this experiment, we adopted the search space and training mechanism as in the Zen-NAS work (Lin et al., 2021). All NAS methods are different only in the accuracy prediction scores used. Table 1 ….
>
> In the experiments for results in Table 1, we adopted the search space and training mechanism as in the Zen-NAS work. All NAS methods are different only in the accuracy prediction scores used. Therefore, we can be sure that, at least in the experiment of NAS for CIFAR10/100, the TPC score is the deciding factor of the superior performance of TPC-NAS.
>
> 4.	the mutation operation in algorithm 2 is not defined, and in the main text there is no reference to the search space defined in the appendix.
> Response:
> Thank you very much for your comments. The mutation operation used is described in the revised manuscript.  Part of the first paragraph of Sec. 3.5 is revised
>
> …architecture in the search space. The MUTATE function creates new
> models from the search space according to Algorithm 3.  ….
>
> Also, a new Algorithm 3 is added to the main text. In addition, we have noted in the main text that all search spaces of the considered application are described in Appendix A.1. The first paragraph of Sec. 3.5 is revised as
>
> …. algorithm is described in Algorithm 2. In Appendix A.1, we have delineated in more detail the search spaces adopted in three different applications. ….

---

> > ### Comment · Reviewer_3zBH · 2022-12-13
> > **Thank you for your response**
> >
> > I thank the authors for their response. The updates certainly improve the clarity of the paper, however based on the results I would still rather keep my original score.

---

### Decision · Program_Chairs · 2023-01-20

**Decision:**

Reject

**Justification For Why Not Higher Score:**

Criticisms by reviewers included novelty, clarity, comparability of methods, and the lack of ablation studies. All reviewers give rejection scores.

**Justification For Why Not Lower Score:**

N/A

**Metareview: Summary, Strengths And Weaknesses:**

This paper introduces a new zero-cost proxy for NAS: the total path count (TPC) and uses it with an evolutionary algorithm to find strong architectures for image classification, object detection and super resolution.
While these applications are interesting, the reviewers prominently criticized the evaluation of the core contribution of TPC on just 20 architectures from the Zen-NAS paper. Other points of criticism included novelty, clarity, comparability of methods, and the lack of ablation studies.
Due to these points all reviewers' scores for this paper are marginally below the acceptance threshold, and I follow the reviewers' suggestion.
I would, however, like to strongly encourage the authors to continue this work and to strengthen the empirical assessment of the core TPC proxy on common benchmarks. A potentially easy way to do this on many benchmarks could be to use the contemporary work on NAS-Bench-Suite Zero that was just presented in the NeurIPS datasets and benchmark track (https://openreview.net/forum?id=yWhuIjIjH8k), which contains the performance of 13 zero cost proxies on 28 tasks. This benchmark may allow a quick way to scientifically study for which types of NAS benchmarks TPC works well and for which it does not work, as well as to understand when one should use it over other zero cost proxies.